# The SnaFab versus the Razi antivenom for treatment of snakebite envenomation: A randomized, double-blind (investigator and victims), active controlled, non-inferiority clinical trial

Seyed Reza Mousavi[1], Alihasan Rahmani[2], Maryam Amini Pouya[3], Elnaz Zabihi Eidgahi[4], Mohamad Delirrad[5], Hamed Hosseini[6], Alireza Ghassemi Toussi[1], Behnaz Hedayatjoo[7,8], Seyyedeh Maryam Afshani[9], Mohammad Amin Ghobadi[9,10], Mohammad Reza Shahidi[9], Seyed Amirhossein Mousavi[1], Maryam Barghbani[11], Mahdi Jannati Yazdan Abad[12]*

**1** Medical Toxicology Research Center, Mashhad University of Medical Sciences, Mashhad, Iran, **2** Department of Medical Toxicology and Forensic Medicine, Faculty of Medicine, Ahvaz Jundishapur University of Medical Sciences, Ahvaz, Iran, **3** Department of Pharmaceutics, Faculty of Pharmacy, Tehran University of Medical Sciences, Tehran, Iran, **4** Department of Pharmacy Practice, Shahid Beheshti University of Medical Sciences, Tehran, Iran, **5** Department of Medical Toxicology and Forensic Medicine, Faculty of Medicine, Urmia, Iran, **6** Medical Faculty, Tehran University of Medical Sciences, Tehran, Iran, **7** Science and Research Branch, Islamic Azad University, Tehran, Iran, **8** Medical department Arta Pharmed Company, Tehran, Iran, **9** Faculty of Pharmacy, Tehran University of Medical Sciences, Tehran, Iran, **10** Vaccine Research Center, Nivad Pharmed Salamat Company, Tehran, Iran, **11** University of Sao Paulo, Ribeirao Preto School of Medicine (FMRP), Ribeirao Preto, Brazil, **12** Department of Clinical Pharmacy, School of Pharmacy, Mashhad University of Medical Sciences, Mashhad, Iran

* Jannatim901@gmail.com

## Abstract

Snakebite envenomation is a significant public health issue in Iran, and it is crucial to have effective and easily accessible antivenom treatments. This study aimed to evaluate the non-inferiority of the SnaFab and the Razi antivenoms for treating snakebite envenomation. The study was a randomized, double-blind, multicenter, active-controlled, non-inferiority trial involving 98 snakebite victims. The patients received either the SnaFab or the Razi antivenoms, with the primary endpoint being the recovery rate within 48 hours after antivenom infusion. The secondary endpoints included adverse events over a 14-day monitoring period and total antivenom consumption. The non-inferiority margin (NIM) was set at 20 percent. The study was commenced on April 17th, 2020, and concluded on May 15th, 2021. The recovery rate was 100 percent in the SnaFab group, which was non-inferior to the Razi antivenom group (98%). The mean number of antivenom units administered was 8.95 ± 4.40 units in the SnaFab group, compared to 9.04 ± 4.49 units in the Razi group (P value: 0.92). Adverse events were reported by 10.2% of individuals in the SnaFab group and 20.4% in the Razi group (p value = 0.27). Muscle weakness was the most commonly reported adverse event in the Razi group (8%), while vertigo was most

**Data availability statement:** All relevant data are within the paper and its Supporting Information files.

**Funding:** This research was funded by Padra Serum Alborz Company (grant number 99/20880). Mohammad Amin Ghobadi is employed by Padra Serum Alborz Company, the manufacturer of SnaFab antivenom. The article processing charges (APC) were also funded by Padra Serum Alborz Company. The funders had no role in study design, data collection and analysis, decision to publish, or preparation of the manuscript.

**Competing interests:** I have read the journal's policy and the authors of this manuscript have the following competing interests: Mohammad Amin Ghobadi is employed by Padra Serum Alborz Company, the manufacturer of SnaFab antivenom. Dr. Hamed Hosseini, who served as Head of Statistical Analysis, is affiliated with Tehran University of Medical Sciences and has no financial relationship with Padra Serum Alborz Company. All other authors declare that no competing interests exist.

**Abbreviations:** NIM, Non-inferiority trial; WHO, World Health Organization; DSMB, data safety and monitoring board; IFDA, Iranian Food and Drug Administration; CRO, Contract Research Organization; INR, International Normalization Ratio; PT, Prothrombin Time; PTT, Partial Thrombin Time; ECG, Electrocardiogram; cm, Centimeter; PP, Per-Protocol; CI, Confidence Interval; SOC, System Organ Class; HAQ, Health-care Access and Quality

frequent in the SnaFab group (4%). Moreover, there were no reports of anaphylactic shock or serum sickness. In summary, this study found that SnaFab antivenom is non-inferior to Razi antivenom for treating snakebite envenomation in Iran. Overall, the incidence of adverse events was comparable between groups, with no statistically significant differences observed.

## Background

Snakebite envenomation represents a significant yet preventable cause of death globally [1], particularly affecting developing nations [2]. The vast majority (approximately 95%) of snakebite cases occur in low and middle-income countries [2], particularly affecting poor and densely populated rural tropical regions [3–5], highlighting a stark healthcare disparity [2]. The World Health Organization's (WHO) stance on snakebite envenomation has evolved over time - initially recognizing it as a neglected tropical disease in 2009, temporarily removing this designation in 2013, before reinstating it as a category A neglected tropical disease in mid-2017 [6]. The organization has established an ambitious target of reducing snakebite-related deaths by 50% by the year 2030 [7]. The global impact of snakebites is substantial, with annual estimates indicating 4.5-5.4 million bite incidents, of which 1.8-2.7 million results in envenomation [2]. Historical data show varying mortality figures: late 1990s estimates suggested over 100,000 annual deaths [8], while subsequent studies in the following decade indicated a range of 20,000–94,000 deaths per year [9]. More recent data from 2019 shows approximately 63,400 deaths (95% uncertainty interval 38,900–78,600), corresponding to an age-standardized mortality rate of 0.8 per 100,000 population. This represents a notable 36% reduction in age-standardized mortality rates since 1990 [1]. According to Iran domestic reports, snakebite is a concerning health issue, affecting 4500–6500 individuals annually [10]. In Iran, snakebites lead to approximately 3–12 deaths annually, which translates to a rate of 0.003 to 0.01 snakebite deaths per 100,000 individuals each year [11].

Based on a recent study by Kazemi et. al, northern to northeastern and western to southwestern areas of Iran contain high richness of venomous snake species [12]. Among 89 different species identified in Iran, of which 30 are venomous, seventeen are mildly-venomous and the remainder are considered non-venomous [13]. As reported by the World Health Organization the venomous snakes in Iran that are of greatest medical significance include the elapid species, specifically the Central Asian cobra (*Naja oxiana*), as well as several Viperids: the saw-scaled viper (*Echis carinatus*), the Levantine viper (*Macrovipera lebetinus*), and the Persian false horned viper (*Pseudocerastes persicus*). The most medically significant venomous snake families, Elapidae and Viperidae, mainly inhabit in western Iran [12].

Snakebite envenomation is a medical emergency and irreversible complications and even death may occur in case of delayed intervention [11,14]. The primary therapeutic approach for snakebite victims remains the administration of a polyvalent antidote. In Iran, the antidote must effectively counteract the venom of six predominant

and dangerous snake species [14]. Early antivenom administration is crucial for optimal patient outcomes, as it significantly impacts recovery rates and reduces complications. Given the severity of snakebite envenomation and the critical importance of timely treatment, the efficacy of antivenom is paramount.

The introduction of SnaFab as a potential therapeutic intervention raises a pivotal clinical question: does it demonstrate non-inferiority compared to the well-established Razi antivenom? This inquiry carries substantial clinical significance. Non-inferiority studies, like the present investigation, seek to establish whether a novel treatment maintains efficacy comparable to the standard therapy within a predefined acceptable margin [15]. If SnaFab achieves this threshold, it may provide additional advantages including improved availability, reduced side effects, and better cost-efficiency, potentially making it the preferred option under specific clinical conditions.

This clinical trial aims to evaluate the non-inferiority of the SnaFab compared to the Razi antivenom in treating snakebite envenoming in Iran. By setting a clear non-inferiority margin, this trial aims to determine whether SnaFab can be considered a non-inferior alternative to the existing standard of care.

### Rationale of design

The world currently grapples with a persistent antivenom shortage that affects both developing [16–19] and developed nations [20]. Resolving this crisis requires establishing economically viable production methods, ensuring appropriate medication use, and implementing standardized treatment protocols across healthcare systems. This comprehensive approach aims to balance manufacturing sustainability with effective clinical practices to address this critical public health challenge [21]. Contemporary antivenom products utilize two main antibody formats: complete IgG molecules or their antigen-binding portions, known as $F(ab')_2$ and Fab fragments [22]. The SnaFab is a polyvalent immune $F(ab')_2$ that derived from clinically relevant snakes native to geographic regions of Iran. Immunoglobulins prepared by caprylic acid fractionation undergo proteolytic digestion to produce fragmented antibodies. Compared to prior antivenom purification methods like ammonium sulfate precipitation, which was time-consuming and less efficient due to protein aggregation [23], caprylic acid refinement is more practical, cost-effective, and causes fewer adverse reactions in humans [24,25]. Moreover, caprylic acid yielded-fragments have high neutralizing activity and offers a rapid and reliable approach to obtain mass quantity of antivenom [26]. Furthermore, caprylic acid precipitation offers technical simplicity by enabling the proteins of interest (i.e., IgG) to be harvested from the supernatant which prevents the precipitate labile antibody from possible damage and degradation [27]. More importantly, caprylic acid is considered to be a viral removal/inactivation step, apparently by virtue of its ability to disrupt the integrity of enveloped viruses [28,29].

## Methods

### Ethics statement

The study was conducted according to the guidelines of the Declaration of Helsinki, and approved by the Institutional Review Board (or Ethics Committee) of Mashhad University of Medical Sciences (protocol code IR.MUMS.REC.1398.295 and date of approval 2020/01/18).

**Trial design.** This was a randomized, double-blind (investigator and victim), active-controlled, parallel non-inferiority trial of the antivenom produced by Padra Alborz Serum Company (SnaFab) compared to the standard antivenom produced by the Razi Vaccine and Serum Research Institute for treating snakebite envenomation. It was conducted at Imam Reza Hospital in Mashhad, Ayatollah Taleghani Hospital in Urmia, and Razi Hospital in Ahvaz. The data safety and monitoring board (DSMB) independently supervised the trial regarding the study continuation, suspension, withdrawal, and/or termination. The trial was registered at the Iranian Registration of Clinical Trials (IRCT20180515039672N2, https://irct.behdasht.gov.ir/trial/41983).

Per Iranian FDA regulatory mandate (documented in the CTA approval letter, March 14, 2020), the first 15 participants randomized to SnaFab constituted a safety run-in phase under independent DSMB oversight. These participants were

assessed for predefined safety endpoints before the remainder of enrollment continued. The DSMB's recommendation to proceed was confirmed in a follow-up IFDA letter dated August 9, 2020. All participants — including the safety run-in subgroup — were treated and monitored using identical eligibility criteria, intervention protocols, safety assessments, and follow-up schedules. They were prospectively included in the intention-to-treat efficacy and safety analyses for their assigned randomization arms. This protocol adaptation was implemented under ethics approval and was consistent with the U.S. FDA's March 18, 2020 guidance [30] on the conduct of clinical trials during the COVID-19 pandemic, which allows for necessary protocol modifications to ensure participant safety and trial feasibility in public health emergencies.

**Participants.** Participants aged 2–60 years who had a history of snakebite, required antivenom based on the severity scale (Table 1), presented within 12 hours of the bite, and provided *written informed consent* (obtained from the parent/guardian of each participant under 18 years of age, adhering the situation in urgent patients settings) were included to the study and victims who had a history of horse serum allergy, previous snakebite or scorpion sting with antivenom administration, multiple bites, pre-hospital wound manipulation, life-threatening bleeding, marine snakebite, mechanical ventilation requirement, anticoagulant usage, coagulation disorders, significant comorbidities (cardiac, neuromuscular, renal, hepatic), pregnancy or lactation, or prior antivenom administration before reaching the hospital were excluded from the study.

**Interventions.** Antivenom dosing was determined based on the snakebite severity scale (Table 1). Victims were given 5-valent antivenom in Urmia and Ahvaz covering vipers and 6-valent antivenom in Mashhad due to the existence of both vipers and cobras. According to the snakebite severity scale (Table 1), all victims with moderate and severe symptoms were prescribed 5 and 10 vials of antivenom, respectively. Cobra bite victims, which might have occurred in Imam Reza Hospital in Mashhad, who did not exhibit clinical signs and symptoms (mild symptoms) were prescribed five vials of 6-valent antivenoms. All prescribed vials were diluted in sodium chloride 0.9% and infused intravenously. The victims were closely monitored 30 minutes, 1, 6, 12, 24, 48, and 72 hours after the initial administration. In situations where INR was more than 3, PT more than 20 seconds, PTT more than 50 seconds or platelets less than 25,000 (six hours after administration of the initial dose), or bite site swelling was progressed, systemic symptoms were continued, or neurotoxicity, and cardiovascular symptoms were worsened (one hour after administration of the initial dose), five antivenoms were re-administered. Maintenance dosing should commence upon reaching therapeutic response (cessation of swelling, improvement in neurological symptoms, and resolution of coagulation panel) or after receiving 20 vials of antivenom (whichever comes first). Two vials of antivenom were diluted into 200 ml of sodium chloride 0.9% and infused every 6 hours for three consecutive doses. All adverse events were carefully recorded during the study. The delayed complications (Serum Sickness) were evaluated up to 14 days after the first antivenom infusion.

The study followed the standard protocol (S1 Table), with all medical care and supportive treatments carefully recorded.

**Outcome.** The primary objective was to determine the proportion of victims who had recovered within 48 hours post-antivenom infusion. Recovery was defined as meeting at least 3 of the following 6 criteria within 48 hours post-antivenom infusion: cessation of swelling progression, PT normalization (<20 seconds) [if abnormal at baseline], INR normalization (<1.2) [if abnormal at baseline], PTT normalization (<50 seconds) [if abnormal at baseline], platelet count recovery (>150,000/μL) [if abnormal at baseline], cessation of neurotoxicity progression.

Secondary outcomes included the proportion of patients who experienced adverse events and the total amount of antivenom administration.

**Sample size.** Sample sizes of 42 in Razi and 42 in SnaFab arm achieve 81% power to detect a non-inferiority margin difference between the group proportions of -0.2000. The reference group proportion is 0.9000. The treatment group proportion is assumed to be 0.7000 under the null hypothesis of inferiority. The power was computed for the case when the actual treatment group proportion is 0.9000. The test statistic used is the one-sided Score test (Farrington & Manning). The significance level of the test was targeted at 0.0250. The significance level actually achieved by this design is 0.0260. Assuming a 15% dropout rate, therefore, 98 victims were participated in the trial.

**Table 1. Snakebite severity classification in clinical trial of SnaFab vs. Razi antivenom in snakebite victims [14,57,58].**

| Severity | Viper bite | Cobra bite |
|---|---|---|
| Mild | **Local:** local swelling of less than 2.5 cm around the bite site<br>**Systemic:** None<br>**Number of antivenom needed:** None (Excluded from the study) | **Local**: None<br>**Systemic:** None<br>**Number of antivenom needed:** 5 vials of 6-valent |
| Moderate | **Local:**<br>Complete swelling of each finger, local swelling of more than 2.5 cm around the bite site, progressive swelling, considerable pain, tissue destruction (skin or muscle necrosis), blister formation, cervical bite, regional lymphadenitis<br>**Systemic:**<br>Intractable vomiting, General weakness, Spontaneous hemorrhage, Cardiac dysrhythmia and abnormal ECG, Increase in serum creatinine, Thrombocytopenia (platelet count less than 150 thousand per cubic millimeter), Increasing coagulation tests (International Normalized Ratio (INR) more than 1.2, PT (Prothrombine Time) more than 20 seconds, PTT (Partial Prothrombine Time) more than 50 seconds)<br>**Number of antivenom needed:** 5 vials of 5-valent | **Local:**<br>Any amount of swelling, pain of any intensity, tissue destruction (skin or muscle necrosis), blister formation, cervical bite, regional lymphadenitis<br>**Systemic:**<br>Ptosis, Fasciculation, Paralysis of the external muscles of the eye, Limb paralysis, Diplopia<br>**Number of antivenom needed:** 5 vials of 6-valent |
| Severe | **Local:**<br>Cervical bite with the possibility of upper airway obstruction, Compartment syndrome<br>**Systemic:**<br>Cardiovascular collapse (cardiac arrest, overt shock [SBP<80 mmHg and decreased peripheral perfusion]), Rhabdomyolysis, Severe active bleeding<br>**Number of antivenom needed:** 10 vials of 5-valent | **Local:**<br>Cervical bite with the possibility of upper airway obstruction, Compartment syndrome<br>**Systemic:**<br>Medulla paralysis, Respiratory paralysis, Rhabdomyolysis<br>**Number of antivenom needed:** 10 vials of 6-valent |

**Randomization.** A computer-generated block randomization method was used in the trial (https://www.sealedenvelope.com/). The participants were randomly allocated to receive either the SnaFab or the control antivenom (Razi antivenom) in a 1:1 ratio after the eligibility criteria were met and informed consent was obtained. The randomization sequence was concealed by an independent contract research organization (CRO). Neither the participants nor the investigators knew which intervention groups the participants were assigned.

**Blinding.** In the efficacy study, all victims were examined by a physician at the study site after assessing their eligibility for inclusion in the study. Once the intervention group for each victim was determined using a pre-prepared randomization chain, the responsible nurse prepared the infusion bag using the study site's drug stock and administered the intravenous infusion. Given the identical appearance of the infusion bags and the administration process, none of the victims were aware of their treatment group. Additionally, efforts were made to ensure that the treating physician was as uninformed as possible about the antivenom group received. This was emphasized in the training provided to the nurses and the physician. Finally, since the study information recorded in the electronic Case Report Form was provided to the data management team in the form of codes lacking identifying information about the victims, the blinding process was fully implemented at the subject level and the study result analysis team, and as much as possible, it was also implemented at the treating physician level in a relative manner

**Statistical methods.** The primary analysis was based on a per-protocol (PP) approach, which included only patients who completed the study without significant protocol deviations. The non-inferiority margin (NIM) was set at 0.20, and the

difference in treatment response rates between the SnaFab and Razi antivenom groups would be calculated with a 95% confidence interval (CI).

Both intent-to-treat (ITT) and per-protocol (PP) analyses were planned and conducted. Since all randomized participants (n = 98) completed the study without major protocol deviations, the ITT and PP populations were identical, and the results presented represent both analysis approaches.

Secondary endpoints, such as the total number of administered antivenom and adverse event rates, were analyzed using appropriate statistical tests (e.g., chi-square test for categorical variables, t-test, or Mann-Whitney U test for continuous variables) with significance set at $p < 0.05$. All analyses were conducted using "SPSS".

## Results

Recruitment for the study commenced on April 17th, 2020, and concluded on May 15th, 2021. 147 individuals were screened, with only 98 eligible (Fig 1) due to exclusion criteria. These criteria encompassed factors such as age range (<2–60 < years), non-snake related complaints, prior antivenom treatment, bites exceeding 12 hours, low bite intensity scores, and those who opted not to participate. All 98 randomized participants completed the study protocol without major deviations. Therefore, the intent-to-treat and per-protocol analysis populations were identical, strengthening the

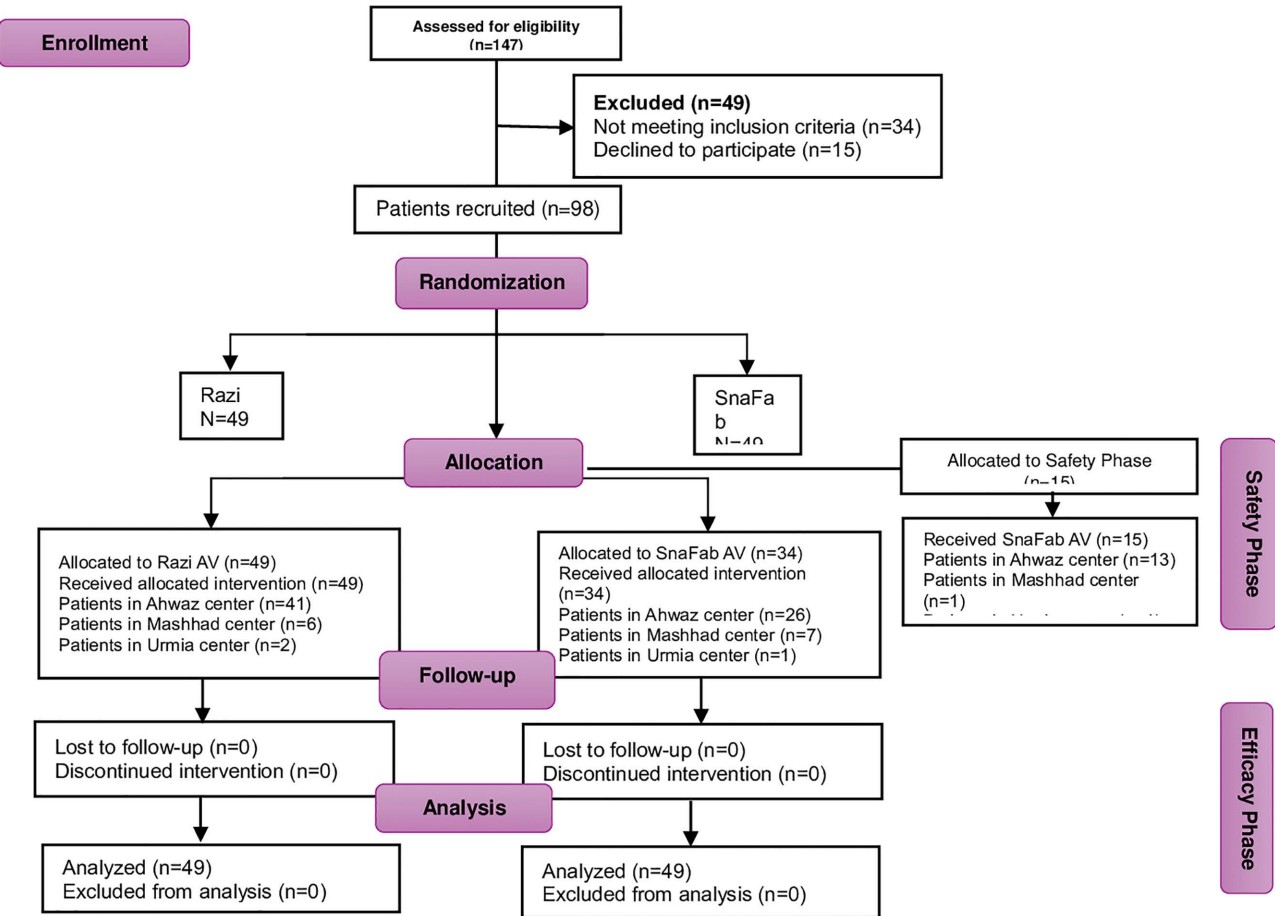

**Fig 1. Consort Chart of Phase III clinical trial of SnaFab compare to Razi antivenom in snakebites.**

robustness of our findings. A total of 34 participants were deemed ineligible for the study based on the following exclusion criteria:

• Age Constraints (n=5):

Participants outside the specified age range of 2–60 years

• Inappropriate Complaint Type (n=4):

Individuals with non-snake bite related complaints

• Pre-hospital Antivenom Administration (n=16):

Participants who received antivenom prior to hospital admission

• Delayed Presentation (n=7):

Patients presenting more than 12 hours after snake bite incident

• Severity Assessment (n=2)

Individuals determined not to require antivenom based on bite severity scale

• Additionally, 15 potential participants declined to provide informed consent for study participation.

Enrolled participants received treatment and were carefully monitored for immediate responses or adverse effects. The primary efficacy endpoint was evaluated within 48 hours after antivenom infusion, with follow-up visits scheduled at 24, 48, and 72 hours and weekly for the first month post-enrollment. These intervals were selected to ensure a thorough comparison of the two antivenoms and to confidently establish the non-inferiority of the SnaFab compared to the Razi antivenom. The study aimed to evaluate both the antivenom treatments' short- and long-term effects to provide dependable and accurate results.

Baseline characteristics are presented in Table 2. 41 (83.67%) participants in the SnaFab group and 37 (75.51%) participants in the Razi serum group were male. The average age of the participants in the SnaFab arm was $34.90 \pm 11.82$ years, and the Razi arm was $28.67 \pm 12.73$ years (p value = 0.01).

Forty-eight out of 49 participants in the Razi arm (98%) and 49 out of 49 participants in the SnaFab arm (100%) showed recovery within 48 hours after antivenom infusion, not exceeding the non-inferiority margin (Fig 2). The difference in recovery rates (SnaFab - Razi) was 0.02 (95% CI: -0.02 to 0.06), demonstrating non-inferiority of SnaFab to Razi antivenom as the confidence interval lies entirely above the pre-specified non-inferiority margin of -20%.

All patients in both SnaFab and Razi group presented with local symptoms including pain and swelling. Upon admission, 35 patients (71.4%) in the Razi group and 26 patients (53.1%) in the SnaFab group exhibited systemic manifestations in addition to local symptoms, including coagulation disorders, thrombocytopenia, and general weakness (Table 2). Severe symptoms, such as active severe hemorrhage and compartment syndrome, were observed in 3 patients (6.12%) from the SnaFab group and 2 patients (4.08%) from the Razi group. Additionally, according to the snakebite severity scale (Table 1), moderate symptoms were documented in 46 patients (93.8%) from the SnaFab group and 47 patients (95.9%) from the Razi group (Table 2).

**Secondary endpoints.** According to the study, the average amount of antivenom administered to victims in the SnaFab group ($8.95 \pm 4.40$ units) and the Razi group ($9.04 \pm 4.49$ units) did not show a significant difference (P value: 0.92). Of the 93 patients presenting with moderate envenomation symptoms according to the snakebite severity scale, 46 patients were treated in the SnaFab group and 47 patients in the Razi group. The total antivenom consumption was 394 vials in the SnaFab group and 402 vials in the Razi group (P value: 0.71). Among the study population, severe envenomation was documented in 5 cases at presentation. The SnaFab cohort included three such cases (patients #1,

**Table 2. Demographic information and clinical presentation of victims in Phase III non-inferiority clinical trial of SnaFab compare to Razi antivenom in snakebites.**

| Characteristic | Overall n = 98 | SnaFab n = 49 | *Razi* n = 49 |
|---|---|---|---|
| Sex | | | |
| Female (%) | | 8 (16.4%) | 12 (24.5%) |
| Male (%) | | 41 (83.6%) | 37 (75.5%) |
| Age [Years] (SD) | | 34.9 ± 11.82 | 28.67 ± 12.73 |
| Bite distribution | | Left lower limb: 19 | Left lower limb: 17 |
| | | Right lower limb: 12 | Right lower limb: 19 |
| | | Left upper limb: 5 | Left upper limb: 8 |
| | | Right upper limb: 12 | Right upper limb: 5 |
| | | Neck: 1 | Neck: none |
| Local: n* (%, total reactions) | | Swelling>2.5 cm: 36 (46.1%, 78) | Swelling>2.5 cm: 42 (53.9%, 78) |
| | | Significant pain: 38 (48.1%, 79) | Significant pain: 41 (51.9%, 79) |
| | | Progressive swelling: 7 (77.8%, 9) | Progressive pain: 2 (22.2%, 9) |
| | | Profound digital swelling: 3 (60.0%, 5) | Profound digital swelling: 2 (40.0%, 5) |
| | | Blister formation: 3 (75.0%, 4) | Blister formation: 1 (25.0%, 4) |
| | | Pain (any severity): 3 (42.9%, 7) | Pain (any severity): 4 (57.1, 7) |
| | | Tissue necrosis (dermal, muscle): 1 (100.0%, 1) | Tissue necrosis (dermal, muscle): 0 (0.0%, 1) |
| Systemic: n* (%, total reactions) | | Coagulopathy: 13 (50.0%, 26) | Coagulopathy: 13 (50.0%, 26) |
| | | Active bleeding: 1 (50.0%, 2) | Active bleeding: 1 (50.0%, 2) |
| | | Thrombocytopenia: 5 (50.0%, 10) | Thrombocytopenia: 5 (50.0%, 10) |
| | | Generalized malaise: 5 (41.7%, 12) | Generalized malaise: 7 (58.3%, 12) |
| | | Compartment syndrome: 1 (100.0%, 1) | Compartment syndrome: 0 (0.0%, 1) |
| | | Spontaneous bleeding: 2 (50.0%, 4) | Spontaneous bleeding: 2 (50.0%, 4) |
| | | Intractable vomiting: 3 (42.9%, 7) | Intractable vomiting: 4 (57.1%, 7) |
| | | Generalized pain: 0 (0.0%, 1) | Generalized pain: 1 (100.0%, 1) |

*n = number of patients developing specific complications; % = percentage of patients in each group; total = cumulative number of all local or systemic reactions (patients may have multiple symptoms simultaneously).

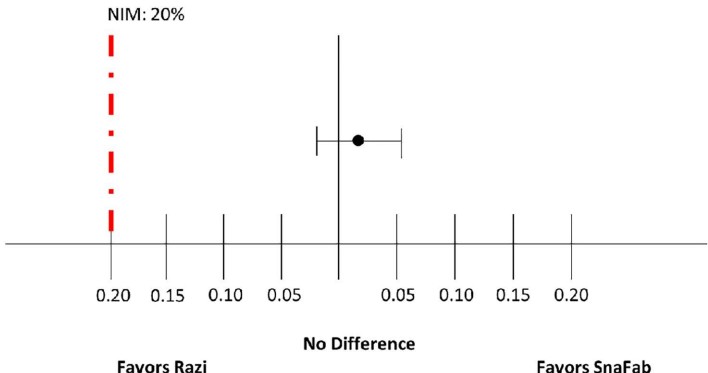

**Fig 2. The non-inferiority margin in Phase III clinical trial of SnaFab compared to Razi antivenom in snakebites.**

#4, and #44) requiring 20, 15, and 10 vials respectively, while the Razi cohort included two cases (patients #50 and #63) requiring 10 and 15 vials respectively. The difference in antivenom requirements between groups was not statistically significant (P = 0.59).

The adverse events reported during the follow-up period are presented in Table 3. During the follow-up period, 10 out of 49 individuals (20.40%) in the Razi group reported adverse events, while 5 out of 49 individuals (10.20%) in the SnaFab group reported adverse events (p value = 0.27). However, there were no cases of anaphylactic shock reported in either group. The Razi and SnaFab group reported muscle weakness and vertigo as the most common adverse event, respectively. No delayed hypersensitivity reactions, including serum sickness, were reported in either group.

## Discussion

This randomized clinical trial provides initial evidence that the SnaFab antivenom is non-inferior to the standard Razi antivenom for treating snakebite victims in Iran. SnaFab antivenom was also well-tolerated and safe.

The predominant demographic of snakebite victims in our study were adult males, which aligns with global trends documented in multiple studies [13,31,32]. Based on a recent systematic review and meta-analysis on global morbidity and mortality of snakebite, among 57 of 65 of included studies 60% of victims were male [33]. This gender disparity in snakebite incidents is often attributed to the increased outdoor activity levels among males, particularly in younger age groups, as they engage more frequently in agricultural and recreational activities that place them at higher risk for snake encounters [31,34]. In terms of severity, 95% of victims reported as moderate symptoms and remaining individuals developed severe symptoms. Regarding the concern about the limited number of severe envenomation cases (n = 5) in our study, this distribution actually aligns with established epidemiological patterns in snakebite severity. According to Afroz et. al, systematic review and meta-analysis study on snakebite morbidity and mortality, the natural distribution of snakebite severity typically follows a pattern where approximately: 46.7% of cases are mild, 28.2% are moderate and roughly 9.9% present as severe envenomation [33]. Our study's relatively small number of severe cases (n = 5) therefore reflects this expected

**Table 3. Adverse events during follow-up period in Phase III non-inferiority clinical trial of SnaFab compared to Razi antivenom in snakebites.**

| System Organ Class (SOC) Preferred Term | SnaFab n (%) | Razi n (%) |
|---|---|---|
| Central & peripheral nervous system disorders | | |
| Vertigo | 2 (4.1) | 3 (6.1) |
| Body as a whole-general disorder | | |
| Malaise | 1 (2.0) | 1 (2.0) |
| Fever | 0 (0.0) | 1 (2.0) |
| Headache | 1 (2.0) | 0 (0.0) |
| Musculo-skeletal system disorders | | |
| Muscle Weakness | 0 (0.0) | 4 (8.2) |
| Skin and appendages disorders | | |
| Dermatitis | 1 (2.0) | 2 (4.1) |
| Bite Site Necrosis | 1 (2.0) | 1 (2.0) |
| Resistance mechanism disorders | | |
| Infection Bacterial | 1 (2.0) | 0 (0.0) |
| Secondary terms-events | | |
| Bite Site Pain | 1 (2.0) | 1 (2.0) |
| Bite Site Swelling | 0 (0.0) | 1 (2.0) |

PLOS Global Public Health

epidemiological distribution rather than a methodological limitation. This natural stratification of severity is consistently observed in clinical settings, suggesting our sample is representative of real-world snakebite presentations.

The primary outcome analysis demonstrates that the SnaFab does not exceed the NIM compared to the Razi antivenom. The trial suggests that the SnaFab can appropriately neutralize venom toxins and halt the progression of envenoming. In addition, the similar total antivenom usage provides further evidence for the non-inferiority of the SnaFab and the Razi antivenoms in neutralizing both viper and cobra venoms.

During the follow-up period, 10 out of 49 individuals (20.40%) in the Razi group reported adverse events, while 5 out of 49 individuals (10.20%) in the SnaFab group reported adverse events. However, there were no cases of anaphylactic shock reported in either group. The Razi and SnaFab arm reported muscle weakness and vertigo as the most common adverse event, respectively. No delayed hypersensitivity reactions, including serum sickness, were reported in either group. The incidence of these adverse events would be compared to Abubakar S. et. al, that evaluate non-inferiority of ET-Plus and ET-G antivenom, reporting ET-Plus caused one or more reactions in 50/194 (25.8%) compared to 39/206 (18.9%) for ET-G antivenom [16]. Moreover, early antivenom reactions were 15.2% and 57% with 10 cc and 20 cc of EchiTAB-Fab antivenom [35] and 17% SAVP Echis antivenom [36].

Our findings highlight the importance of post-administration monitoring for adverse events following antivenom treatment. The 14-day follow-up period in our study was prospectively designed with the primary aim of identifying early and delayed adverse effects, particularly the onset of serum sickness, a well-recognized systemic hypersensitivity reaction following the administration of heterologous proteins such as those found in snake antivenoms [37]. Early and delayed hypersensitivity reactions can vary depending on factors such as antivenom type, purification method, foreign protein amount, and composition (whole immunoglobulin or Fab fragments) [38]. The pathophysiology of serum sickness involves the formation of circulating antigen-antibody immune complexes, typically 5–14 days after exposure to the foreign protein (in this case, the antivenom). These immune complexes can become trapped in the endothelium of small blood vessels in various tissues, including the skin, joints, and kidneys, leading to complement activation, an influx of inflammatory cells, and subsequent tissue damage [39–41]. This immunological cascade manifests clinically with symptoms such as fever, rash (often urticarial or maculopapular), arthralgia, myalgia, lymphadenopathy, and occasionally proteinuria [23,39]. It is worth noting that our study did not involve any prophylaxis protocol, and none of the patients experienced anaphylactic reactions.

The classic description by von Pirquet and Schick in 1905 indicated that serum sickness symptoms typically manifest between 5–20 days post-administration of foreign serum [42]. More contemporary reviews and clinical observations refine this, noting that while the range can be 1–22 days, the majority of serum sickness cases associated with antivenom therapy present within 5–14 days after administration [4,23]. The incidence of serum sickness following antivenom therapy varies widely, reported from <5% to over 70% in different studies, influenced by factors such as the type and purity of the antivenom (e.g., whole IgG vs. F(ab')2 or Fab fragments), the dose administered, the animal source of the immunoglobulins, and individual patient factors [23,43]. For example, older, less purified antivenoms are generally associated with higher rates of serum sickness [23]. Certain antivenoms have a reaction frequency as high as 25% [44–46].

Several antivenom safety studies have employed similar follow-up periods with demonstrated effectiveness: Dart et al. conducted a comprehensive safety evaluation of crotalid antivenom (CroFab) in their study, employing systematic follow-up protocols to monitor for delayed reactions. Their study of 11 patients demonstrated the effectiveness of structured post-treatment monitoring. In no subject did an allergic reaction develop [47,48]. Similarly, Lavonas et al. in their evidence-informed consensus workshop on crotaline snakebite management, established follow-up protocols that emphasized monitoring during the critical 5–7-day period for serum sickness development [49]. Warrell's extensive work on antivenom safety has consistently shown that the majority of clinically significant delayed antivenom reactions occur within the first two weeks' post-administration [4]. Isbister & Brown (2012) employed systematic 7–10-day follow-up protocols in their randomized controlled trial of Latrodectus antivenom, providing valuable methodology for comprehensive safety assessment [50].

Given this well-established onset window, a 14-day follow-up period is a common and rational approach in clinical trials evaluating antivenoms to capture the peak incidence and majority of serum sickness cases. Many clinical studies assessing antivenom safety and efficacy employ follow-up schedules that include assessment around day 7 and day 14 (or extend up to 21 days) specifically to monitor for delayed adverse reactions like serum sickness. Therefore, our 14-day follow-up window was strategically chosen as it is well-suited for capturing the development of serum sickness in most patients who would experience this immunologically-mediated reaction, aligning with the typical timeframe for immune complex formation and symptomatic presentation. This allows for timely identification and management of this important delayed adverse event.

Fortunately, none of the patients in our study exhibited fever, malaise, or an itchy rash after two weeks, so they did not need to revisit the hospital for further evaluation. Given the absence of severe adverse events and the similar profiles of other adverse effects, the two antivenoms have a comparable safety profile (p value = 0.27). Conversely, Abubakar S. et. al, reported severe reactions including bronchospasm, gastrointestinal symptoms more frequently in ET-Plus than ET-G (10.8% compared to 5.3%, respectively). The authors debated that severe reactions may probably reflect the higher dose of anti-venom administration. More importantly, Abubakar S. et. al, reported only one quarter of the patients for 2-week follow-up, and among them late serum sickness reported 10.2% and 5.2% in ET-Plus and ET-G, respectively [16]. In African Antivipmyn, 19% of patients developed unexpected events including shock, dyspnoea, cough and angioedema [51].

The low incidence of minor and severe adverse events may be attributed to caprylic acid fractionation. Non-Ig-G proteins including albumin precipitate in refinement process and maintain negligible detection of contaminants and reflected by lower adverse effects in different trials [45,52]

The average amount of antivenom in both groups showed no significant differences (P value = 0.92). It suggests both antivenom require similar dosing regimens and may imply comparable potency between products. Moreover, the interchangeability of products is supported. These data are particularly valuable for: healthcare policy makers, clinical protocol development, hospital formulary decisions, resource allocation planning and cost-effectiveness analyses.

The lack of standardized treatment protocols for snakebite is a significant concern. Different antivenoms may have different dosing and administration requirements, and there is currently no consensus on the best approach to treating snakebite envenomation; due to the need for a single protocol to carry out the project, project researchers managed to write an evidenced-based snakebite protocol in Iran with the help of existing papers.

The limited availability of antivenom in some areas is a matter of great concern. In a comprehensive study of the global mapping of snake bites by Longbottom et al. in 2018, the authors provide a map showing nearly 300 snake species globally. Nearly 6.9 billion people reside within a range of snake-inhabited areas, and unfortunately, about 146.70 million linger within far-distant regions lacking appropriate quality health-care provisioning. In this study, patients are divided into ten deciles (HAQ: Health-care Access and Quality) based on quality and access to the urban center (population over 50 thousand people) for treatment measures and access to antivenom, where one is the least accessible and ten is the most accessible. It is estimated that more than 10 million people in Iran are exposed to snakebite (one-eighth of the country's population). More than 341 thousand people are in the seventh decile of access to treatment measures (without considering access to antivenom). Besides, it is estimated that 272.91 million individuals (65.25%) of the inhabitants within the lowest decile (lowest HAQ) are at grave risk of exposure to any snake for which no effective therapy, including antivenom availability [53]. Antivenoms were manufactured for 119 (43%) of 278 snake species investigated by WHO [53].

The findings of our non-inferiority trial comparing Razi and SnaFab antivenoms have significant implications that align with WHO's "Snakebite Envenoming: A Strategy for Prevention and Control" roadmap, which aims to reduce snakebite mortality and disability by 50% before 2030 through improved access to safe, effective, and affordable antivenoms [54]. By demonstrating non-inferiority of SnaFab to Razi, our study supports WHO's goal of strengthening regional antivenom manufacturing capacity [55], as having multiple validated manufacturers within Iran enhances supply security and reduces dependence on imported products. The availability of two comparable antivenoms could foster healthy market

competition, potentially leading to more affordable pricing, thereby addressing WHO's emphasis on antivenom accessibility and affordability [17]. Furthermore, our rigorous clinical trial methodology provides evidence-based validation of antivenom quality, supporting WHO's focus on ensuring effective and safe products reach patients [56]. This study exemplifies WHO's recommendation for developing context-specific solutions, as both antivenoms are specifically designed for Iranian snake species and healthcare settings [53]. Having multiple validated manufacturers strengthens the sustainability of antivenom supply, addressing WHO's concern about ensuring consistent availability of these essential medicines. These findings contribute to the broader goal of improving global antivenom accessibility while maintaining high standards of safety and efficacy, directly supporting WHO's strategic objectives for reducing the burden of snakebite envenoming worldwide [54].

## Limitation

Pandemic-related: Implementing the trial coincided with the COVID-19 pandemic, which took longer than expected to reach the desired sample size.

Regulatory mandated safety run-in: a key design feature was the regulatory-mandated safety run-in phase. The first 15 randomized SnaFab participants were assessed by the DSMB before full-scale recruitment; they were retained in the efficacy analysis to preserve statistical power, consistent with "seamless" trial designs accepted in rare and high-risk conditions. While such a design may raise concerns about potential heterogeneity, this risk was minimal because (i) eligibility criteria were identical across groups, and (ii) treatment protocols and monitoring were unchanged. This design adaptation was driven by regulatory requirements and unprecedented recruitment limitations during the COVID-19 pandemic, consistent with FDA guidance [30] which supported necessary adaptive measures to protect participants and preserve trial integrity during the pandemic. Nonetheless, because recruitment occurred in two stages, some readers may interpret this as a sequential design; we have clarified the protocol in the Methods and provided regulatory documentation to avoid misinterpretation.

Methodological-related: Most participants were male, which may reflect the higher risk of snakebite among males due to their increased exposure to outdoor activities. However, it is essential to note that gender differences were not statistically significant between the two groups, indicating that the randomization process successfully created comparable groups regarding gender distribution. The SnaFab group's mean age was significantly higher than the Razi group. While this difference may be a potential confounding factor, it is unlikely to impact the study results significantly.

## Conclusions

Recent research conducted in Iran has revealed that the SnaFab antivenom is non-inferior to the Razi antivenom in treating snakebites. Furthermore, both antivenoms were generally well tolerated, with no significant differences in the incidence of adverse events between groups. These findings emphasize the importance of geographically tailored antivenom solutions and continued innovation in snakebite therapeutics. It is crucial to conduct post-marketing surveillance to monitor rare adverse events and evaluate efficacy to ensure the safety of patients.

### Consent for publication

Written Informed consent was obtained from all subjects and parent/guardian of each participant under 18 years of age involved in the study.

### Supporting information

**S1 Table. Supportive treatments and other necessary medical care in clinical trial of SnaFab vs. Razi antivenom in snakebite victims.**
(DOCX)

**S1 Protocol.  Study Protocol.**
(DOCX)

**S1 Checklist.  CONSORT Checklist.**
(DOC)

**S1 Data.  Data.**
(DOCX)

## Acknowledgments

We would like to thank our colleagues and peers for their support and encouragement throughout the process of this research. We also appreciate the constructive feedback from anonymous reviewers, which has helped improve the quality of this paper.

## Author contributions

**Conceptualization:** Seyed Reza Mousavi.

**Data curation:** Seyed Reza Mousavi, Maryam Amini Pouya, Hamed Hosseini, Maryam Barghbani.

**Formal analysis:** Hamed Hosseini, Seyyedeh Maryam Afshani.

**Funding acquisition:** Mohammad Amin Ghobadi, Mohammad Reza Shahidi.

**Investigation:** Seyed Reza Mousavi, Alihasan Rahmani, Mohammad Amin Ghobadi, Mohammad Reza Shahidi, Maryam Barghbani, Mahdi Jannati Yazdan Abad.

**Methodology:** Hamed Hosseini.

**Project administration:** Seyed Reza Mousavi.

**Resources:** Mohammad Amin Ghobadi, Mohammad Reza Shahidi.

**Software:** Hamed Hosseini.

**Supervision:** Seyed Reza Mousavi, Maryam Amini Pouya.

**Validation:** Seyed Reza Mousavi, Alihasan Rahmani, Elnaz Zabihi Eidgahi, Mohamad Delirrad, Alireza Ghassemi Toussi, Behnaz Hedayatjoo.

**Writing – original draft:** Seyed Reza Mousavi, Mahdi Jannati Yazdan Abad.

**Writing – review & editing:** Seyed Reza Mousavi, Maryam Amini Pouya, Hamed Hosseini, Seyed Amirhossein Mousavi, Mahdi Jannati Yazdan Abad.

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
