## [Decision Letter · Decision Letter 0]

11 Apr 2025

PGPH-D-25-00087

The SnaFab™ Versus The Razi Antivenom for Treatment of Snakebite Envenomation: A Randomized, Double-Blind (investigator and victims), Active Controlled, Non-inferiority Clinical Trial

Dear Dr. Mahdi Jannati Yazdan Abad,

Thank you for submitting your manuscript to PLOS Global Public Health. After careful consideration, we feel that it has merit but does not fully meet PLOS Global Public Health’s publication criteria as it currently stands. Therefore, we invite you to submit a revised version of the manuscript that addresses the points raised during the review process.

We look forward to receiving your revised manuscript.

Kind regards,

Muhammed O Afolabi, MD, MPH, PhD

Academic Editor

Journal Requirements:

1. We do not publish any copyright or trademark symbols that usually accompany proprietary names, eg (R), (C), or TM (e.g. next to drug or reagent names). Please remove all instances of trademark/copyright symbols throughout the text, including ® on page 3 and ™ on the title page.

2. In the online submission form, you indicated that [The datasets generated during and/or analyzed during the current study are available from the corresponding author on reasonable request.].

a. In a public repository,

b. Within the manuscript itself, or

c. Uploaded as supplementary information.

3. Graphical Abstract.pdf: Please confirm whether you drew the images / clip-art within the figure panels by hand. If you did not draw the images, please provide (a) a link to the source of the images or icons and their license / terms of use; or (b) written permission from the copyright holder to publish the images or icons under our CC-BY 4.0 license. Alternatively, you may replace the images with open source alternatives. See these open source resources you may use to replace images / clip-art:

- https://openclipart.org/

Additional Editor Comments :

The authors reported the findings of a clinical trial evaluating SnaFab™ as a safe, non-inferior alternative to Razi antivenom. The randomized, double-blind, active-controlled non-inferiority trial design minimizes bias and strengthens validity. The outcomes are clear with the primary (48-hour recovery rate) and secondary endpoints (adverse events, antivenom consumption) are well-defined and clinically relevant. SnaFab™ demonstrated non-inferiority to Razi antivenom (100% vs. 98% recovery rate) with a predefined non-inferiority margin (20%) and had fewer adverse events (10.2% vs. 20.4%) and no severe reactions (anaphylaxis/serum sickness).

Despite the methodological rigor and innovative purification method involving the use of caprylic acid fractionation for SnaFab™ which likely contributed to reduced adverse events, I have some concerns summarised below:

1. Sample Size and Recruitment: High exclusion rate (49/147 screened), raising concerns about selection bias. Also, small subgroup with severe envenomation (n=5), limited conclusions for critical cases. Gender imbalance (male-dominated cohort) reflects real-world exposure but reduces generalizability to females.

2. Confounding Factors: Significant age difference between groups (SnaFab™: 34.9 vs. Razi: 28.7 years; P-value not reported). Regional focus (Iran) limits extrapolation to other geographic settings with different snake species.

3. Analysis Limitations: Per-Protocol Analysis excludes dropouts, risking overestimation of efficacy. Intention-to-treat analysis is missing. Authors should consider including intention-to-treat analysis to complement per-protocol results. They should also adjust for age imbalance using multivariate regression.

4. Short Follow-Up: 14-day monitoring may miss delayed adverse effects (e.g., chronic sequelae). Authors should provide information on the extended follow-up to to assess delayed complications (e.g., serum sickness, chronic joint issues).

5. Safety Reporting: Adverse events were self-reported or clinician-observed, potentially underestimating mild/moderate reactions. Authors should specify criteria for "recovery" (e.g. quantitative measures of swelling/coagulation). Authors provided no details on blinding success (e.g., whether treating physicians remained unaware of allocation). Authors should report measures taken to ensure blinding integrity (e.g., surveys of investigators/participants).

6. Conflict of Interest: Funding and employment ties to Padra Serum Alborz (manufacturer of SnaFab™) introduce potential bias, despite rigorous methodology. Authors should disclose funder involvement in study design/data analysis

7. Global Relevance: Authors should discuss implications of their trial findings in relationship with global relevance for WHO’s snakebite roadmap and alignment with global antivenom accessibility goals.

Reviewers' comments:

Reviewer's Responses to Questions

**Comments to the Author**

1. Does this manuscript meet PLOS Global Public Health’s publication criteria?

Reviewer #1: Yes

2. Has the statistical analysis been performed appropriately and rigorously?

Reviewer #1: Yes

3. Have the authors made all data underlying the findings in their manuscript fully available (please refer to the Data Availability Statement at the start of the manuscript PDF file)?

Reviewer #1: No

4. Is the manuscript presented in an intelligible fashion and written in standard English?

Reviewer #1: No

Reviewer #1: The authors present results from a non-inferiority trial of SnaFab to Razi, two antivenom treatments, among snake bite victims in Iran. Recovery within 48 hours was assessed. Adverse events (AE) are also summarized. Authors found similar rates of recovery in the two arms, falling within the non-inferiority margin. Fewer individuals in the SnaFab group experienced adverse events than in the Razi group, though this difference was not statistically significant. The manuscript will be strengthened if the authors consider the following points.

1. In the abstract, results and discussion, authors should provide a fair comparison of the two groups in relation to adverse events. Specifically, they highlight the most commonly reported AE in the Razi group, but do not provide the most common AE in the SnaFab group. There is also a paragraph in the discussion about the low incidence of AE in the SnaFab group, but the percentage of individuals experiencing an AE in the two groups was not significantly different. The Conclusion similarly states that SnaFab boasts a more favorable safety profile. That is too strong of a statement given their data.

2. In the definition of recovery (the primary outcome), authors state that at least 3 recovery criteria had to be met - the coagulation abnormalities recovery includes 4 possibilities, so would each of those count? Also, I am assuming an individual had to have abnormal coagulation in order for the coagulation components to be considered in the recovery - is that correct? If so, how would someone without coagulation abnormalities meet the recovery criteria, since there are only 2 other aspects of recovery that could be met?

3. In the sample size section, authors should include the base proportion, as they do in the protocol. As stated in the manuscript, there is insufficient information to replicate the power calculation.

4. The non-inferiority margin is 20% which seems rather large when comparing treatments to something as potentially dangerous as snakebites. Was there any rationale behind this choice of the NIM?

5. Authors list reasons for exclusion in the 1st paragraph in the results, but they should provide how many were excluded for each reason (also the exclusion related to age would be outside the age range, not 2-60 years).

6. Authors need to refer to Figure 2 in the text. The title for Figure 2 should have "compare to" changed to "compared to". Also, I believe authors have flipped the Favors SnaFab and Favors Razi in the figure. Authors are reporting SnaFab-Razi, so the right of no difference would be SnaFab having higher recovery than Razi.

7. Authors are encouraged to carefully read through the manuscript. There are missing words (such as "Based on recent study..." on page 2 instead of "Based on a recent study"), extra capitalization ("and Irreversible complications" also on page 2), and other grammatical or typographical errors. Another example is "relevant snake's native to geographic regions of Iran" on page 3..."snake's" should be "snakes".

Minor points:

1. the word "data" is plural, so authors should use "data show" or "data ...show" (first paragraph in Background section). Similarly, "These data are" should be used instead of "This data is" (page 10, 2nd paragraph)

2. In the 1st paragraph of the Background, authors refer to an "uncertainty range" - is this a 95% confidence interval?

3. on page 3, 2nd line, authors refer to frequency of snakebites in a sentence about early antivenom administration. I'm not sure how the frequency of snakebites is related to the timing of administration. I think the authors are combining two different ideas here and should clarify the sentence.

4. page 3, in the Rationale of design section, the sentence "Immunoglobins prepared..." is an incomplete sentence and should be revised.

5. Table 1: an asterisk is used after shock in the description of Severe for viper bites, but the asterisk is not defined.

6. Since the protocol states that intent-to-treat analysis will also be performed, authors should state somewhere that the analyses conducted were both per protocol and intent-to-treat (since everyone who was randomized completed the study without deviations).

7. The enrollment dates are given in the Methods section of the Abstract, but the results section of the main manuscript. For consistency, they should be reported in the same section of the Abstract and main text.

8. Figure 1 - what is the "allocated to the safety phase" box for? Authors state that all participants were evaluated for safety throughout the follow-up, so I'm not sure what is different about those 15. This should be clarified.

9. Table 2: in the title, change "compare to" to "compared to". Also, it is not clear to me in the Local (and possibly systemic) section, what n (%, total) represents...in the Local section, the total is more than the number of participants for some local symptoms. Authors should clarify what is being reported.

10. page 7, %98 and %100 should be changed to 98% and 100%. Also, authors should remove "and not exceeding the NIM" and talk about that when referring to Figure 2.

11. Some of the information in the Secondary Endpoints section of the results was already reported in the prior paragraph (# with moderate symptoms in each group and # with severe symptoms in each group). In the prior paragraph, authors refer to Table 2 after a sentence on the number in each group with moderate symptoms, but Table 2 does not easily identify moderate symptoms, so referring to that table does not make sense.

**Do you want your identity to be public for this peer review?** For information about this choice, including consent withdrawal, please see our Privacy Policy

Reviewer #1: No

---

## [Decision Letter · Decision Letter 1]

1 Jul 2025

PGPH-D-25-00087R1

The SnaFab™ Versus The Razi Antivenom for Treatment of Snakebite Envenomation: A Randomized, Double-Blind (investigator and victims), Active Controlled, Non-inferiority Clinical Trial

Dear Dr.  YazdanAbad,

Thank you for submitting your revised manuscript to PLOS Global Public Health. After careful consideration, we feel that it has merit but still does not fully meet PLOS Global Public Health’s publication criteria as it currently stands. Therefore, we invite you to submit a further revised version of the manuscript that addresses the points raised during the review process.

We look forward to receiving your revised manuscript.

Kind regards,

Muhammed O Afolabi, MD, MPH, PhD

Academic Editor

Journal Requirements:

Reviewers' comments:

Reviewer's Responses to Questions

**Comments to the Author**

Reviewer #1: (No Response)

publication criteria?

Reviewer #1: Yes

3. Has the statistical analysis been performed appropriately and rigorously?

Reviewer #1: Yes

4. Have the authors made all data underlying the findings in their manuscript fully available (please refer to the Data Availability Statement at the start of the manuscript PDF file)?

Reviewer #1: Yes

5. Is the manuscript presented in an intelligible fashion and written in standard English?

Reviewer #1: Yes

Reviewer #1: The authors addressed the majority of my earlier comments. However, their response raised some new questions and added text had some typographical errors that should be addressed. Specifically, authors should address the following:

1. In the author response to my prior question regarding the "allocated to safety phase" box, authors state that there was a sequential design that began with 15 patients, all receiving SnaFab to assess safety prior to the start of the efficacy/non-inferiority trial, but that those 15 patients were ultimately included in the SnaFab group for the trial. In Figure 1, authors show that participants were randomized and then there were the 15 SnaFab assessed for safety, which does not seem to match the sequential design described in their response. The sequential design would suggest those first 15 were not actually randomized, but instead were just given SnaFab. Authors need to describe their design within the manuscript and ensure that Figure 1 matches that design.

Minor points:

1. In a few places, authors have the "%" sign before the number (for example, twice in the Abstract (%8 and %4) and %60 and %95 in the Discussion) - authors should ensure when they report percentages, the "%" sign follows the number.

2. Table 2 has the incorrect percentage for the SnaFab group for male (41/49=83.67% as stated in the text, not 86.7% in the table. Also, Table 2 has a superscript "1" above each of the sample sizes, but I'm not sure what that refers to (there is no note corresponding to a superscript 1). Also, the font for front right limb is different from the rest of the table.

**Do you want your identity to be public for this peer review?** For information about this choice, including consent withdrawal, please see our Privacy Policy

Reviewer #1: No

---

## [Editor Report · Decision Letter 2]

7 Oct 2025

The SnaFab™ Versus The Razi Antivenom for Treatment of Snakebite Envenomation: A Randomized, Double-Blind (investigator and victims), Active Controlled, Non-inferiority Clinical Trial

PGPH-D-25-00087R2

Dear Dr Mahdi Jannati YazdanAbad,

We are pleased to inform you that your manuscript 'The SnaFab™ Versus The Razi Antivenom for Treatment of Snakebite Envenomation: A Randomized, Double-Blind (investigator and victims), Active Controlled, Non-inferiority Clinical Trial' has been provisionally accepted for publication in PLOS Global Public Health.

Best regards,

Muhammed O Afolabi, MD, MPH, PhD

Academic Editor